# A Large Multicenter Brazilian Case-Control Study Exploring Genetic Variations in Interferon Regulatory Factor 6 and the Risk of Nonsyndromic Cleft Lip With or Without Cleft Palate

**DOI:** 10.3390/ijms26073441

**Published:** 2025-04-07

**Authors:** Renato Assis Machado, Daniella Reis Barbosa Martelli, Silvia Regina de Almeida Reis, Luiz Evaristo Ricci Volpato, Rafaela Scariot, Juliana Feltrin-Souza, Ana Lúcia Carrinho Ayroza Rangel, Hercílio Martelli-Júnior, Ricardo D. Coletta

**Affiliations:** 1Department of Oral Diagnosis and Graduate Program in Oral Biology, School of Dentistry, University of Campinas, Piracicaba 13414-018, São Paulo, Brazil; renatoassismachado@yahoo.com.br; 2Master Program, School of Dentistry, Ingá University Center, Maringá 87035-510, Paraná, Brazil; 3Primary Care/Health Sciences Postgraduate Program, State University of Montes Claros, Montes Claros 39401-089, Minas Gerais, Brazil; daniellareismartelli@yahoo.com.br (D.R.B.M.); hercilio.junior@unimontes.br (H.M.-J.); 4Department of Basic Science, Bahiana School of Medicine and Public Health, Salvador 40290-000, Bahia, Brazil; srareis@uol.com.br; 5Postgraduate Program in Integrated Dental Sciences, Cuiaba School of Dentistry, University of Cuiaba, Cuiaba 78000-000, Mato Gosso, Brazil; lemcvolpato@uol.com.br; 6Department of Stomatology, Federal University of Paraná, Curitiba 80120-170, Parana, Brazil; rafaela_scariot@yahoo.com.br (R.S.); julianafeltrin1@gmail.com (J.F.-S.); 7Center of Biological Sciences and of the Health, School of Dentistry, State University of Western Paraná, Cascavel 85819-110, Paraná, Brazil; alrangel2002@yahoo.com.br; 8Center for Rehabilitation of Craniofacial Anomalies, Dental School, University of José Rosario Vellano (Unifenas), Alfenas 37130-000, Minas Gerais, Brazil; 9Oral Pathology and Oral Medicine, Dental School, State University of Montes Claros, Montes Claros 39401-089, Minas Gerais, Brazil

**Keywords:** IRF6, nonsyndromic orofacial clefts, single-nucleotide polymorphism, case-control study, risk factor

## Abstract

Nonsyndromic cleft lip with or without cleft palate (NSCL ± P) is strongly associated with both environmental and genetic risk factors, but its genetic underpinnings remain partially known. While variants in interferon regulatory factor 6 (IRF6) are linked to NSCL ± P risk in populations from Asia and Europe, studies on the highly admixed Brazilian population are scarce and have produced ambiguous results. This study aimed to investigate the contribution of *IRF6* variants to the risk of NSCL ± P. Five tag-single nucleotide polymorphisms (rs599021, rs2073485, rs2235375, rs7552506, and rs642961) were analyzed in a large multicenter cohort composed of 1006 patients with NSCL ± P and 942 healthy controls. Statistical analyses involved multiple logistic regression tests consideration the tri-hybrid genetic origin of the Brazilian population, under a Bonferroni *p* value correcting for multiple comparisons. The A allele (OR: 1.43, 95% CI: 1.22–1.67, *p* < 0.0001) and AA genotype (OR: 2.04, 95% CI: 1.46–2.86, *p* < 0.0001) frequencies of rs642961 were significantly associated with NSCL ± P risk. Stratified analyses indicated that the variant is associated with susceptibility to both nonsyndromic cleft lip only (NSCLO) and nonsyndromic cleft lip and palate (NSCLP). However, the association with NSCLO was primarily observed in patients with high African ancestry, whereas the association with NSCLP was predominantly seen in patients with high European ancestry. No significant associations were found for the other investigated variants. Our results support the notion that the *IRF6* rs642961 variant represents a marker of susceptibility to NSCL ± P in the Brazilian population, and that genetic ancestry composition plays a central role in the association with the cleft type.

## 1. Introduction

Orofacial cleft, the most prevalent malformation affecting the face, results from fusion failures of the facial processes during early embryogenesis. Depending on the process involved in the failure, it is classified as cleft lip only (CLO), involving lack of fusion between the medial nasal and maxillary processes, cleft palate only (CPO), represented by non-fusion of the palatal shelves, which develop as bilateral outgrowths from the maxillary processes, or cleft lip and palate (CLP), due to nonunion of all these facial processes [1]. The similarities in both epidemiologic features and embryologic timing allows the combination of CLO and CLP in a unique group, called cleft lip with or without cleft palate (CL ± P). The majority of cases (approximately 70% of cases of CL ± P and 50% of cases of CPO) are isolated and sporadic in occurrence and classified as nonsyndromic, whereas the presence of rare genetic mutations characterizes the syndromic forms of orofacial clefts [2]. The incidence of nonsyndromic orofacial clefts is influenced by diverse factors, such as ethnicity, environmental exposures and socioeconomic status, and ranges from 1:500 to 1:2500 newborns [3].

The etiopathogenesis of nonsyndromic orofacial clefts is credited to a complex interplay of genetic, epigenetic, and environmental factors [4]. In recent decades, a plethora of studies, adopting varied genetic approaches, have identified diverse genes and loci of susceptibility to nonsyndromic orofacial clefts. Among the first described genes was interferon regulatory factor 6 (IRF6). Historically, *IRF6* was targeted for investigation in nonsyndromic CL ± P (NSCL ± P) after mutations were detected in patients with Van der Woude syndrome, a Mendelian disorder that may include orofacial cleft in its clinical spectrum. The first study revealed an association of the genetic polymorphic variant rs2235371 with NSCL ± P [5], but a later study by the same group demonstrated the association of rs2235371 with NSCL ± P dependents on rs642961, a single nucleotide polymorphism (SNP) that abrogates the transcription factor AP-2a binding site in the promoter of *IRF6* [6]. The association of rs642961 with NSCL ± P has been frequently replicated in Asian and European populations [7,8,9,10], but it was not confirmed in studies involving populations from Africa [11,12], Iran [13], Yemen [14] and Honduras [15], or in Hispanic and non-Hispanic white individuals living in the United States [16]. The association of *IRF6* with NSCL ± P in the Brazilian population remains uncertain [17,18,19,20,21]. Together these findings provide evidence of the influence of ancestral origin on *IRF6* risk, and highlight the importance of identifying both *IRF6* common and population-specific candidate clinical variants of NSCL ± P. Accordingly, a recent study demonstrated that the *IRF6* variant rs570516915, which is highly enriched in the Finnish population, is a risk factor for NSCPO [22].

Brazil, with a population surpassing 210 million, has one of the most genetically heterogeneous populations in the world, which is the result of more than 500 years of interethnic admixture between Native Americans, European colonizers, and African slaves [23]. The clinical influence of the genomic profile of the individual in the susceptibility of common and rare diseases has been demonstrated in different situations [24], including that of NSCL ± P [25]. The aim of the current study was to investigate the potential association of *IRF6* genetic variants, selected by the tag-SNP strategy and represented by rs599021, rs2073485, rs2235375, rs7552506, and rs642961, with NSCL ± P in a large multicenter Brazilian case-control cohort.

## 2. Results

This study included 1006 patients with NSCL ± P, with 273 affected by NSCLO and 733 affected by NSCLP, and 942 healthy controls. There were no significant differences in genomic ancestry between the NSCL ± P group and controls. However, the NSCL ± P group included more men (56.7% vs. 46.5%; *p* < 0.0001) than the control group, and this difference was driven by the high proportion of males with NSCLP (Table 1). The genotyping call rate ranged from 98.4% to 99.7%, and the concordance in the validation sample, which was genotyped twice, was 100%. The genotype distributions of all variants adhered to HWE (Table 2).

Multiple logistic regression analyses considering the differences in sex and genomic ancestry proportions were used to compare allele and genotype distributions between the NSCL ± P and control groups. Although the SNP rs599021 has demonstrated some associations with a nominal *p* value (*p* < 0.05), only rs642961 was found to be associated with NSCL ± P risk, after adopting a p value for multiple comparisons of ≤0.01. The other SNPs (rs2073485, rs2235275, and rs7552506) showed no significant associations with NSCL ± P susceptibility (Table 3). Power analysis showed very good statistical power to detect association with the current sample size for all SNPs. The only variant that did not reach >99% was rs2235375, with a power of 84.6%.

The presence of the rs642961 A allele was associated with an increased risk of NSCL ± P (OR:1.43, 95% CI: 1.22–1.67, *p* < 0.0001). Patients carrying the AA genotype exhibited a higher risk of NSCL ± P (OR: 2.04, 95% CI: 1.46–2.86, *p* < 0.0001), and both dominant (OR: 1.35, 95% CI: 1.12–1.64, *p* = 0.002) and recessive models (OR: 1.96, 95% CI: 1.40–2.72, *p* < 0.0001) further indicated an increased risk for NSCL ± P (Table 3). Next, we calculated the population attributable fraction (PAF), representing the proportion of NSCL ± P due to rs642961 A allele. The PAF for heterozygous carriers (GA genotype) was 7.47%, and for homozygous carriers (AA genotype) it was 25.92%.

To further explore the allele and genotype contribution of *IRF6* variants to NSCL ± P susceptibility, a stratified analysis considering type of cleft was performed. The findings indicate that the A allele of rs642961 was associated with a significantly higher susceptibility to NSCLO (OR: 1.40, 95% CI: 1.11–1.77, *p* = 0.003) and NSCLP (OR: 1.44, 95% CI: 1.21–1.70, *p* < 0.0001) compared to the G allele (Table 4). For NSCLO, the AA genotype (OR: 1.92, 95% CI: 1.18–3.11, *p* = 0.01) and the recessive genetic model (GG + GA vs. AA) also showed a significant association (OR: 1.81, 95% CI: 1.13–2.91, *p* = 0.01), whereas the higher susceptibility to NSCLP was confirmed with the AA genotype (OR: 2.09, 95% CI: 1.46–2.99, *p* < 0.0001) and in the dominant (OR: 1.35, 95% CI: 1.09–1.66, *p* = 0.004) and recessive (OR: 2.01, 95% CI: 1.41–2.86, *p* < 0.0001) genetic models (Table 4).

To strengthen our understanding of the contribution of rs642961 to NSCL ± P risk, ancestry-stratified analysis was performed, revealing noteworthy and specific associations for rs642961 (Table 5). For this analysis, aligning with the pioneer strategy proposed by Pena and collaborators [26], which has been successfully adopted by us in previous studies [27,28], individuals whose genome demonstrated a predominance of more than 20% African genetic markers were designated as having a high African ancestral background. Conversely, individuals whose genome exhibited high European markers and less than 20% of African markers were classified as possessing a high European ancestral background. The present results revealed that the association of the A allele (OR: 1.35, 95% CI: 1.14–1.61, *p* = 0.0006) with increased risk of NSCL ± P in patients exhibiting high European ancestry is due to the frequency of this allele in patients with NSCLP (OR: 1.40, 95% CI: 1.16–1.70, *p* = 0.0004). On the other hand, the susceptibility of NSCL ± P (OR: 1.79, 95% CI: 1.25–2.56, *p* = 0.001) in patients with high African ancestry is related to the predominance of the variant allele in NSCLO (OR: 2.32, 95% CI: 1.44–3.74, *p* = 0.0005).

## 3. Discussion

Although the mechanisms leading to the disruption of the embryological processes resulting in nonsyndromic orofacial clefts are complex and multifactorial, our understanding of the genetic components is evolving. To date, more than 50 loci/genes have been associated with nonsyndromic orofacial clefts, with many being validated in independent studies on diverse populations [29]. The validation of candidate variants for nonsyndromic orofacial clefts is essential, particularly in admixed populations such as the Brazilian population, because this allows an accurate estimation of the influence of population structure, avoiding the misinterpretation of risk scores calculated for other populations. After the identification that the common *IRF6* rs642961 variant is related to increased odds for NSCL ± P, several studies explored its susceptibility and that of other *IRF6* SNPs in the Brazilian population. The different genotyping methods, the small sample sizes, and the lack of control over the inference of the ancestry structure of the samples produced inconsistent and contradictory results. After combining the tag-SNP strategy, a large and multicenter sample and statistical analyses controlling individual ancestry structure, the results of the current study revealed that the presence of the rs642961 variant A allele increased the risk of NSCL ± P in the admixed Brazilian population.

The first Brazilian study exploring the association of *IRF6* variants with NSCL ± P risk was reported by Paranaiba and colleagues in 2010 [17]. This study examined the variants rs2235371 and rs642961 in 177 samples of NSCL ± P and 126 controls by restriction fragment length polymorphism analysis, and revealed a lack of involvement of these *IRF6* polymorphisms with NSCL ± P. Brito et al. [18] found that the A allele of rs642961 was not associated with NSCL ± P, but stratified analysis by cleft type showed a significant association with NSCLO, but not NSCLP. Interestingly, the authors discussed that the association was probably driven by the patients with cleft from a specific region of Brazil, where heritability was estimated to have an important influence on NSCL ± P risk. However, the patients with NSCLO in this cohort showed a higher African ancestry contribution than individuals of the control group. In a sample derived from subjects born in Bahia state, the Brazilian state with the highest historical immigration of Africans and a population that has the highest proportion of African genomic ancestry in Brazil, do Rego Borges et al. [19] observed that the frequency of the A risk allele of rs642961 was higher in the NSCL ± P group compared to the control group, but the significance did not withstand Bonferroni correction for multiple tests. The study conducted by Souza et al. [20], exploring 259 triads with patients with nonsyndromic orofacial clefts and their parents, which has the advantage of minimizing influence differences in the population stratification within and between groups, revealed that rs642961 is not associated with NSCL ± P in the Brazilian population, though the haplotype composed of the G allele of rs2235771 and the A allele of rs642961 was overtransmitted to patients when all types of clefts (NSCLO, NSCPO, and NSCLP) were combined. Bezerra et al. [21] observed no association between rs642961 and NSCL ± P, but the minor allele of rs2235771, which is frequently found in strong disequilibrium of linkage with rs642961, was associated with increased risk for NSCPO, but not NSCL ± P. However, the study included only 38 samples of NSCPO, and the p value was not corrected by multiple comparisons. Altogether, the variability of the results of the previous studies contrasting with the robust results reported here suggest that the lack of power of the samples may have limited the determination of the magnitude of the effects of rs642961 in previous studies. The result of the meta-analysis with these previous studies exploring rs642961 in the Brazilian population reinforces this evidence (Figure 1). The combination of 1089 samples of NSCL ± P and 1038 controls revealed low heterogeneity in the frequencies of the alleles across the individual studies, and yielded a pooled OR of 1.30 (95% CI: 1.08–1.56, *p* = 0.005).

The studies presented here also revealed that the ancestry composition influences the association of rs642961 with the type of cleft. In the sample with high European ancestry, the susceptibility of the A allele was with NSCLP but not with NSCLO, whereas the association in patients with high African ancestry was only with NSCLO. In a population of more than 210 million, the highly heterogeneous and admixed structure constructed over 500 years of marriages from individuals from three main different roots (Amerindians, Europeans, and Africans) represents a factor with direct implications on the modulation of genetic risk factors for nonsyndromic orofacial clefts. However, this is the first study to show that genomic ancestry impacts on the association of a specific genetic variant with the type of cleft. The investigation of potential associations of NSCL ± P susceptibility signals identified by genome-wide studies in the Brazilian population revealed that rs227731 at 17q22, which may have regulatory effects on the *NOG* gene, a potent antagonist of bone morphogenetic proteins (BMP), and rs742071, an intronic variant of the *PAX7* gene at 1p36, are associated with NSCL ± P in the Brazilian population due to the high frequency of the minor alleles in individuals of European ethnic background. On the other hand, rs1873147 at 15q22.2, an SNP located in an intergenic region that contains important sites with transcription activity, is involved in the pathogenesis of NSCL ± P in patients with a high percentage of African ancestry [25]. The variants rs3758249 at 9q22 and rs7078160 at 10q25.3 were reported as susceptibility loci for NSCL ± P in this Brazilian population with high African ancestry [19], which contrasts with a finding of no association of them with NSCL ± P in a Brazilian cohort with high European ancestry [30]. Likewise, the *BMP* rs2761887 polymorphism was only significantly associated with NSCL ± P in the Brazilian population enriched by African ancestry [28]. Collectively, these findings strongly support the notion that the intrinsic heterogeneity of the Brazilian population must be acknowledged in the design and interpretation of studies exploring the genetic basis of NSCL ± P. It is important to consider that the ancestry contribution of the subjects included in this study are in line with population genetic studies in Brazilians reporting most individuals have significant degrees of European and/or African ancestry, while Amerindian ancestry is mainly found in individuals of specific regions of Brazil [24,31].

It is important to recognize that the study has limitations and strengths. The multicenter design, enrolling samples from four of the five macroregions of Brazil, is at the same time a strength and a limitation. This approach brings a better representation of the Brazilian population, but the great variability among the different regions from which the samples were collected, particularly in relation to environment factors, potential gene-gene and gene-environment interactions, and heritability, may have influenced the results and the study was unable to control for these factors. Stratification of the sample is always related to loss of power, but in this particular situation, it did not prevent the identification of significant signals for both NSCLO and NSCLP. The strengths of the study include the use of a large sample size, the application of tag-SNP, which covered the presentative regions of *IRF6*, and a robust statistical approach with control for confounding effects including sex and ancestry proportions, with application of correction for multiple comparison tests.

## 4. Materials and Methods

### 4.1. Study Participants

This multicenter case-control study included patients with NSCL ± P and controls from four of the five Brazilian macroregions, who were recruited in the context of the Brazilian Oral Cleft Group (BOCG), a collaborative consortium committed to study risk factors associated with orofacial clefts. The centers associated with BOCG include the Centrinho do Hospital Santo Antônio das Obras Sociais Irmã Dulce in Salvador, Bahia, situated in the northeast region of Brazil; the Centro Pró-Sorriso at Hospital Alzira Velano, UNIFENAS, in Alfenas, Minas Gerais, located in the southeastern region; the Associação de Portadores de Fissura Lábio Palatal (APOFILAB) in Cascavel, Paraná, and the Centro de Atendimento Integral ao Fissurado Labiopalatal (CAIF) in Curitiba, both situated in the southern region, and the Centro de Reabilitação de Fissuras Orais do Hospital Geral e Maternidade de Cuiabá, located in the central-western region of Brazil. In those centers, all patients were screened to detect and exclude those with signs and symptoms of syndromes, and then only patients with NSCL ± P were included in the current study. The controls consisted of Brazilian individuals without orofacial cleft, family history of orofacial cleft or congenital abnormalities, from the same geographical areas as the cases. Control subjects were not related to the participating patients. The study protocol was approved by the institutional Research Ethics Committee (CAAE: 08452819.0.0000.5418), and all participants or their parents or legal guardians provided written informed consent.

### 4.2. Sample Size Estimation

For estimation of the sample size needed to reach a power of 80%, the Quanto software (version 1.2.4, [32]) was used. The calculation was performed with the following parameters: unmatched case-control study, gene only analysis, additive inheritance model, genetic effect of 1.3, minor allele frequency (MAF) of 0.137 (13.7%), two-sided type I error of 0.05, and Brazilian population risk (prevalence) of 0.00147. The genetic effect and MAF were calculated after a meta-analysis of four previous studies that explored rs642961 (details in the legend of Figure 1), the only *IRF6* variant studied more than once in the Brazilian population [17,18,19,21]. With these parameters, the sample size required was 881 per group. The study included samples from 1006 patients with NSCL ± P and 942 healthy controls.

### 4.3. Genetic Polymorphism Selection

The selection of SNPs was performed using the tag-SNP approach to identify representative polymorphisms in *IRF6*. After configuring the Tagger program [33] to a pairwise algorithm, with a r^2^ ≥ 0.80 and MAF ≥ 0.10, 5 independent blocks exhibited genetic disequilibrium. From these blocks, we carefully selected the SNPs with the highest MAF value, giving preference to those that had undergone previous scrutiny in nonsyndromic orofacial clefts. The primary attributes of each SNP are depicted in Table 2.

### 4.4. SNP Genotyping

Desquamated oral mucosal cells obtained from oral swabs were subjected to genomic DNA extraction using a salt-out protocol previously described [34]. After DNA concentration determination using the NanoDrop™ 2000 spectrophotometer (Thermo Fisher Scientific, Waltham, MA, USA), 5 ng of genomic DNA were subjected to SNP genotyping with TaqMan 5′-exonuclease allelic discrimination assays (Assay-on-Demand, Applied Biosystems, Waltham, MA, USA) in the StepOnePlus real-time platform, following the manufacturer’s directions. For quality control, 10% of the samples were randomly selected and the reaction repeated.

### 4.5. Assessment of Genomic Ancestry

To determine the ancestral lineage of each individual, a panel of 40 biallelic short insertion–deletion polymorphisms (INDELs), previously validated as reliable indicators of ancestry within the Brazilian population [35], was genotyped and integrated into the analysis [36]. The Structure software (version 2.3.4) was employed to unravel the genomic ancestry inherent in each participant. This analytical process involved the application of a K = 3 model, duly considering the tri-hybrid origin that characterizes the Brazilian population.

### 4.6. Statistical Analysis

To evaluate the Hardy–Weinberg equilibrium (HWE) and differences in sex distribution, the chi-square test was applied. Proportions of genomic ancestry across groups were subjected to comparison using the Mann–Whitney test. To conduct multiple logistic regression analyses, the SNPassoc package (version 2.1-2) within the Rstudio program (version 3.5.1) [37] was used. These analyses were performed under unrestricted, dominant, and recessive genetic models. Statistical significance after Bonferroni correction for multiple comparisons was defined as *p* ≤ 0.01.

Power for detecting a *p* value ≤ 0.05 for each variant was calculated in the Quanto software (version 1.2.4, [32]), assuming a prevalence of NSCL ± P in Brazil of 0.00146 [38] and using the most conservative odd ratios (OR) reported in the literature. For rs2235375, the OR was derived from the meta-analysis performed by Golshan-Tafti et al. [39]. The population attributable fraction for rs642961 was calculated using Levin’s formula [40].

## 5. Conclusions

The findings of the current study ultimately provide evidence that the rs642961 polymorphism is associated with a significantly increased risk for NSCL ± P at both the allele and the genotype levels. The stratified analysis revealed that the association with NSCLO was driven by patients with high African ancestry and the association with NSCLP was dependent on patients with high European background. These findings not only enrich our understanding of the genetic basis of NSCL ± P, but also offer opportunities to dissect the roles of different *IRF6* variants in the pathogenesis of this common and distressing congenital disease in admixed populations, such as those from the Latin American countries.

## Figures and Tables

**Figure 1 ijms-26-03441-f001:**
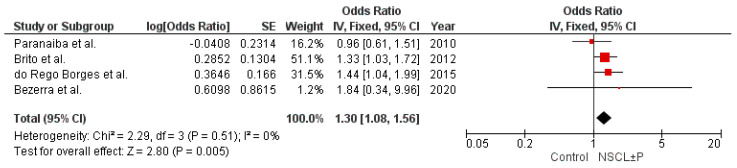
Forest plot for the variant allele distribution of IRF6 rs642961 in the studies with the Brazilian population [17,18,19,21]. The meta-analysis was conducted using the Review Manager (RevMan, version 5.4.1, Cochrane), applying an inverse variance method with fixed effect model (I^2^ < 50%). The square markets indicate the odds ratio of the A allele for different studies, and the size of the marker correlates to the inverse variance of the effect estimate and indicates the weight of the study. The diamond data market indicates the pooled effect.

**Table 1 ijms-26-03441-t001:** Characteristics of the patients included in the study.

	Control (n = 942)	NSCL ± P (n = 1006)	NSCLO (n = 273)	NSCLP (n = 733)
Sex				
Male	438 (46.5%)	570 (56.7%) *	145 (53.1%)	425 (58.0%) *
Female	504 (53.5%)	436 (43.3%)	128 (46.9%)	308 (42.0%)
Ancestry				
European	84.0%	83.3%	82.9%	83.5%
African	14.0%	13.9%	15.0%	13.4%
Amerindian	2.0%	2.8%	2.1%	3.1%

NSCL ± P: nonsyndromic cleft lip with or without cleft palate; NSCLO: nonsyndromic cleft lip only; NSCLP: nonsyndromic cleft lip and palate. * *p* < 0.0001.

**Table 2 ijms-26-03441-t002:** Characteristics of single nucleotide polymorphisms (SNP) in *IRF6*, including genome location, function, major and minor (in bold) alleles, genotyping call rate, and *p* value of the Hardy–Weinberg equilibrium (HWE) test.

SNP	Location	Function	Alleles	Call Rate	HWE
rs599021	chr1: 209787577	3′ UTR	A/**C**	99.2%	0.05
rs2073485	chr1:209789449	Intron	G/**A**	98.4%	0.16
rs2235375	chr1:209792242	Intron	G/**C**	99.2%	0.05
rs7552506	chr1:209796557	Intron	G/**C**	99.7%	0.07
rs642961	chr1:209815925	Promoter	G/**A**	98.5%	0.47

**Table 3 ijms-26-03441-t003:** Associations of *IRF6* variants with the risk of nonsyndromic cleft lip with or without cleft palate (NSCL ± P). The values of *p* were adjusted for covariates (sex and genomic ancestry) by logistic regression analysis.

	Control (%)	NSCL ± P (%)	OR (95% CI)/*p* Value
rs599021			
Allele			
A	62.9	66.1	Reference
C	37.1	33.9	0.87 (0.76–0.99)/0.04
Genotype			
AA	42.1	44.5	Reference
AC	41.7	43.2	0.98 (0.81–1.19)/0.85
CC	16.2	12.4	0.72 (0.55–0.95)/0.02
Dominant (AA/AC + CC)	42.1/57.9	44.5/55.5	1.10 (0.91–1.32)/0.30
Recessive (AA + AC/CC)	83.8/16.2	87.6/12.4	0.73 (0.56–0.94)/0.01
rs2073485			
Allele			
G	77.8	78.1	Reference
A	22.2	21.9	0.98 (0.84–1.14)/0.82
Genotype			
GG	61.3	62.1	Reference
GA	32.9	31.0	0.96 (0.79–1.17)/0.69
AA	5.8	5.9	1.01 (0.68–1.50)/0.95
Dominant (GG vs. GA + AA)	61.3/38.7	62.1/37.9	0.97 (0.80–1.17)/0.73
Recessive (GG + GA vs. AA)	94.3/5.7	94.1/5.9	1.03 (0.70–1.51)/0.89
rs2235275			
Allele			
G	65.0	62.4	Reference
A	35.0	37.6	1.11 (0.97–1.27)/0.10
Genotype			
GG	44.0	40.0	Reference
GC	41.9	44.7	1.17 (0.96–1.43)/0.11
CC	14.1	15.3	1.19 (0.90–1.57)/0.21
Dominant (GG vs. GC + CC)	44.0/56.0	40.0/60.0	1.18 (0.98–1.42)/0.08
Recessive (GG + GC vs. CC)	85.9/14.1	84.7/15.3	1.10 (0.85–1.42)/0.47
rs7552506			
Allele			
G	69.4	69.3	Reference
C	30.6	30.7	1.00 (0.87–1.15)/0.93
Genotype			
GG	50.1	49.4	Reference
GC	38.7	39.9	1.04 (0.86–1.27)/0.66
CC	11.2	10.8	0.97 (0.72–1.32)/0.86
Dominant (GG vs. GC + CC)	50.1/49.9	49.4/50.6	1.03 (0.86–1.23)/0.76
Recessive (GG + GC vs. CC)	88.8/11.2	89.2/10.8	0.96 (0.71–1.28)/0.75
rs642961			
Allele			
G	81.6	75.6	Reference
A	18.4	24.4	1.43 (1.22–1.67)/<0.0001
Genotype			
GG	69.4	62.7	Reference
GA	24.3	25.7	1.17 (0.95–1.45)/0.14
AA	6.3	11.6	2.04 (1.46–2.86)/<0.0001
Dominant (GG vs. GA + AA)	69.4/30.6	62.7/37.3	1.35 (1.12–1.64)/0.002
Recessive (GG + GA vs. AA)	93.7/6.3	88.4/11.6	1.96 (1.40–2.72)/<0.0001

**Table 4 ijms-26-03441-t004:** Distribution of the *IRF6* variants in nonsyndromic cleft lip only (NSCLO) and nonsyndromic cleft lip and palate (NSCLP). The *p* values were adjusted for covariates (sex and genomic ancestry) by logistic regression.

	Control (%)	NSCLO (%)	OR (95% CI)/*p* Value	NSCLP (%)	OR (95% CI)/*p* Value
rs599021					
Allele					
A	62.9	63.7	Reference	66.9	Reference
C	37.1	36.3	0.96 (0.79–1.18)/0.74	33.1	0.83 (0.72–0.97)/0.01
Genotype					
AA	42.1	40.9	Reference	45.8	Reference
AC	41.7	45.6	1.12 (0.84–1.51)/0.43	42.3	0.93 (0.75–1.15)/0.52
CC	16.2	13.5	0.86 (0.56–1.31)/0.47	11.9	0.68 (0.50–0.92)/0.01
Dominant (AA/AC + CC)	42.1/57.9	40.9/59.1	1.05 (0.79–1.39)/0.73	45.8/54.2	1.16 (0.95–1.41)/0.14
Recessive (AA + AC/CC)	83.8/16.2	86.5/13.5	0.81 (0.54–1.20)/0.27	88.1/11.9	0.70 (0.52–0.93)/0.01
rs2073485					
Allele					
G	77.8	81.6	Reference	76.8	Reference
A	22.2	18.4	0.78 (0.61–1.01)/0.06	23.2	1.05 (0.89–1.25)/0.50
Genotype					
GG	61.3	66.8	Reference	60.4	Reference
GA	32.9	29.7	0.83 (0.61–1.12)/0.22	32.9	1.02 (0.82–1.26)/0.88
AA	5.8	3.5	0.56 (0.27–1.17)/0.10	6.7	1.19 (0.79–1.81)/0.40
Dominant (GG vs. GA + AA)	61.3/38.7	66.8/33.2	0.79 (0.59–1.06)/0.10	60.4/39.6	1.04 (0.85–1.27)/0.69
Recessive (GG + GA vs. AA)	94.3/5.7	96.5/3.5	0.60 (0.29–1.23)/0.14	93.3/6.7	1.19 (0.79–1.78)/0.41
rs2235275					
Allele					
G	65.0	63.8	Reference	61.9	Reference
A	35.0	36.2	1.05 (0.85–1.29)/0.61	38.1	1.14 (0.98–1.32)/0.07
Genotype					
GG	44.0	43.8	Reference	38.6	Reference
GC	41.9	39.9	0.96 (0.71–1.29)/0.77	46.5	1.27 (1.02–1.57)/0.03
CC	14.1	16.3	1.16 (0.77–1.74)/0.47	14.9	1.20 (0.89–1.63)/0.23
Dominant (GG vs. GC + CC)	44.0/56.0	43.8/56.2	1.01 (0.76–1.33)/0.95	38.6/61.4	1.25 (1.02–1.53)/0.03
Recessive (GG + GC vs. CC)	85.9/14.1	83.7/16.3	1.19 (0.81–1.73)/0.38	85.1/14.9	1.07 (0.80–1.41)/0.65
rs7552506					
Allele					
G	69.4	71.8	Reference	68.4	Reference
C	30.6	28.2	0.89 (0.71–1.10)/0.29	31.6	1.05 (0.90–1.22)/0.52
Genotype					
GG	50.1	52.9	Reference	48.0	Reference
GC	38.7	37.7	0.92 (0.69–1.24)/0.59	40.6	1.09 (0.89–1.35)/0.48
CC	11.2	9.3	0.79 (0.48–1.28)/0.32	11.4	1.05 (0.76–1.46)/0.63
Dominant (GG vs. GC + CC)	50.1/49.9	52.9/47.1	0.89 (0.68–1.18)/0.41	48.0/52.0	1.08 (0.89–1.32)/0.42
Recessive (GG + GC vs. CC)	88.8/11.2	90.7/9.3	0.81 (0.51–1.30)/0.38	88.7/11.3	1.01 (0.74–1.38)/0.95
rs642961					
Allele					
G	81.6	75.9	Reference	75.5	Reference
A	18.4	24.1	1.40 (1.11–1.77)/0.003	24.5	1.44 (1.21–1.70)/<0.0001
Genotype					
GG	69.4	62.5	Reference	62.8	Reference
GA	24.3	26.6	1.22 (0.88–1.68)/0.23	25.4	1.16 (0.92–1.46)/0.22
AA	6.3	10.8	1.92 (1.18–3.11)/0.01	11.8	2.09 (1.46–2.99)/<0.0001
Dominant (GG vs. GA + AA)	69.4/30.6	62.5/37.5	1.36 (1.02–1.81)/0.03	62.8/37.2	1.35 (1.09–1.66)/0.004
Recessive (GG + GA vs. AA)	93.7/6.3	89.2/10.8	1.81 (1.13–2.91)/0.01	88.2/11.8	2.01 (1.41–2.86)/<0.0001

**Table 5 ijms-26-03441-t005:** Stratified analysis of *IRF6* rs642961 based on ancestry of the samples. Samples were divided in relation to the genomic ancestry in high European genomic ancestry or high African genomic ancestry. The *p* values were adjusted for sex by logistic regression analysis.

	Control (%)	NSCL ± P (%)	OR (95% CI)/*p* Value	NSCLO (%)	OR (95% CI)/*p* Value	NSCLP (%)	OR (95% CI)/*p* Value
High European Ancestry							
Allele							
G	79.9	74.5	Reference	76.4	Reference	73.8	Reference
A	20.1	25.5	1.35 (1.14–1.61)/0.0006	23.6	1.22 (0.93–1.60)/0.14	26.2	1.40 (1.16–1.70)/0.0004
Genotype							
GG	67.3	61.4	Reference	62.8	Reference	60.8	Reference
GA	25.2	26.3	1.14 (0.90–1.46)/0.27	27.2	1.16 (0.80–1.67)/0.43	26.0	1.14 (0.88–1.48)/0.32
AA	7.5	12.3	1.80 (1.25–2.59)/<0.001	9.9	1.42 (0.81–2.49)/0.22	13.2	1.94 (1.32–2.85)/0.0007
Dominant (GG vs. GA + AA)	67.3/32.7	61.4/38.6	1.30 (1.04–1.61)/0.01	62.8/37.2	1.22 (0.87–1.70)/0.25	60.8/39.2	1.32 (1.05–1.67)/0.01
Recessive (GG + GA vs. AA)	92.5/7.5	87.7/12.3	1.73 (1.21–2.48)/0.002	90.1/9.9	1.36 (0.78–2.36)/0.28	86.8/13.2	1.87 (1.28–2.73)/0.001
High African Ancestry							
Allele							
G	87.0	78.9	Reference	74.3	Reference	80.9	Reference
A	13.0	21.1	1.79 (1.25–2.56)/0.001	25.7	2.32 (1.44–3.74)/0.0005	19.1	1.59 (1.07–2.35)/0.02
Genotype							
GG	76.4	67.0	Reference	61.8	Reference	69.1	Reference
GA	21.4	23.9	1.28 (0.82–2.00)/0.28	25.0	1.45 (0.76–2.77)/0.26	23.5	1.21 (0.74–1.98)/0.44
AA	2.3	9.1	4.58 (1.69–12.44)/0.002	13.2	7.20 (2.29–22.61)/0.0007	7.4	3.60 (1.23–10.50)/0.01
Dominant (GG vs. GA + AA)	76.4/23.6	67.0/33.0	1.59 (1.05–2.41)/0.02	61.8/38.2	2.00 (1.12–3.57)/0.02	69.1/30.9	1.44 (0.91–2.28)/0.11
Recessive (GG + GA vs. AA)	97.7/2.3	90.9/9.1	4.32 (1.60–11.67)/<0.001	86.8/13.2	6.56 (2.12–20.32)/0.0008	92.6/7.4	3.44 (1.19–9.97)/0.01

## Data Availability

The data to support the findings of this study will be available on request from the corresponding author.

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
