# Peer review of "A Large Multicenter Brazilian Case-Control Study Exploring Genetic Variations in Interferon Regulatory Factor 6 and the Risk of Nonsyndromic Cleft Lip With or Without Cleft Palate"

_ijms, 2025, doi:10.3390/ijms26073441_

Round 1
Reviewer 1 Report
Comments and Suggestions for Authors
Using a case-control model, the Authors present a study to detect the association between 5 tag SNPs within the IRF6 gene and the risk of nsCL/P in the Brazilian population. The study also took into account the ethnic origin of cases and controls, which allowed secondary analyses to be performed on population strata.
The size of the case study seems quite large. It would be useful, however, for the authors to test the statistical power of the case study about the specific allelic frequencies in the Brazilian population of the variants under study.
It is not clear what criteria the authors used to choose these 5 SNP tags since they were not equally spaced from each other.
In Table 2, the location of 4 out of 5 SNPs refers to the human genome reference GRCh38/hg38. The location of the rs599021 instead seems not to be referring to GRCh38/hg38. According to what is reported in Table 2, the rs599021 locates 145kb downstream of rs642961. According to GRCh38/hg38, rs599021 is located at 209787577, which is 1872 nt upstream of rs2073485. In Table 2, the sequence of SNPs should be reordered.
Tables 3, 4, and 5 are too extensive. Some data could possibly be provided with a supplementary annex. Showing the frequencies of both major and minor alleles is redundant. It would be enough to show only the frequency of minor alleles (MAF).
Regarding frequencies and ORs calculated according to the dominant and recessive models, these could be provided in the text (not in the table) only when there is evidence of significant or borderline-significant p-values ​​of the heterozygous or homozygous genotypes.
Furthermore, it would be useful if the Authors calculated the Population Attributable Fraction (PAF) values to assess the impact of the risk genotypes in the populations.
Author Response
Using a case-control model, the Authors present a study to detect the association between 5 tag SNPs within the IRF6 gene and the risk of nsCL/P in the Brazilian population. The study also took into account the ethnic origin of cases and controls, which allowed secondary analyses to be performed on population strata.
Thank you for highlighting the results of our study, and for the suggestions to improve it.
The size of the case study seems quite large. It would be useful, however, for the authors to test the statistical power of the case study about the specific allelic frequencies in the Brazilian population of the variants under study.
We completely agree that power calculations for negative associations are important to eliminate unreliable results. Following your suggestion, we have performed power analysis of our data using the Quanto software (version 1.2.4, https://pphs.usc.edu/biostatistics-software/#quanto). Power for detecting a p value ≤0.05 for each variant was calculated assuming a prevalence of NSCL±P in Brazil of 0.00146 (Martelli-Junior et al., 2007) and using the most conservative odd ratios (OR) reported in the literature. It is important to mention that there are very few studies with the variants rs2073485, rs7552506 and rs599021 and none involving the Brazilian population. For rs2235375, the OR was derived from the meta-analysis performed by Golshan-Tafti et al. (2024).
Power analysis showed very good statistical power to detect association with the current sample size for all SNPs. The only variant that did not reach >99% was rs2235375, with a power of 84.6%. These data suggest that there may be no association of NSCL±P with these SNPs in this population.
The answer above was included in the revised manuscript, part in results and part in the material and methods.
It is not clear what criteria the authors used to choose these 5 SNP tags since they were not equally spaced from each other.
Thank you for this important question. As originally mentioned, after set the Tagger program to a pairwise algorithm, with a r2≥0.80 and MAF≥0.10, and identification of the independent blocks exhibited genetic disequilibrium, we selected the most representative SNP in each block taking into consideration the highest MAF and the fact of the SNP has undergone previous scrutiny in nonsyndromic orofacial clefts. The position of SNP in the block, respecting similar distribution along the gene, was not taken into consideration.
In Table 2, the location of 4 out of 5 SNPs refers to the human genome reference GRCh38/hg38. The location of the rs599021 instead seems not to be referring to GRCh38/hg38. According to what is reported in Table 2, the rs599021 locates 145kb downstream of rs642961. According to GRCh38/hg38, rs599021 is located at 209787577, which is 1872 nt upstream of rs2073485. In Table 2, the sequence of SNPs should be reordered.
We are sorry for our mistake. The location was corrected in Table 2, and the distribution order of the SNPs was corrected in the text and tables.
Tables 3, 4, and 5 are too extensive. Some data could possibly be provided with a supplementary annex. Showing the frequencies of both major and minor alleles is redundant. It would be enough to show only the frequency of minor alleles (MAF).
We appreciate your comment, and completely agree that MAF may be enough. However, we have opted to display both major and minor allele frequencies allowing a better view of the results for the readers of the study. In the main text, the access to the information is easier than in the supplementary material. Moreover, as an online only article, there is no limit of pages or tables/figures, and space should not be a concern.
Regarding frequencies and ORs calculated according to the dominant and recessive models, these could be provided in the text (not in the table) only when there is evidence of significant or borderline-significant p-values of the heterozygous or homozygous genotypes.
With all due respect, we appreciate your constructive comments, but we have opted to keep the complete analysis, including the different genetic models, in the tables of the main file.
Furthermore, it would be useful if the Authors calculated the Population Attributable Fraction (PAF) values to assess the impact of the risk genotypes in the populations.
Thank you for your astute suggestion. As recommended, we have calculated the population attributable fraction (PAF), representing the proportion of cases that can be attributed to the risk allele, for the variant rs642961 using the classical Levin’s formula. The presence of one copy of the A allele (GA genotype) produced PAF of 7.47%, whereas the PAF for AA genotype was 25.92% in this cohort.
Reviewer 2 Report
Comments and Suggestions for Authors
This manuscript needs some minor changes:
1) It is not clear if p-values in the tables are crude or corrected. Please indicate this.
2) There is no mentioned to results for rs59902 by recessive model (significant p-value) in Results or Discussion.
3) Results within Table 5 are confused because showed association separated by ancestry origin but also adjusted by ancestry. In my opinion the authors must decide to segregate or to adjust.
4) Why the authors did not include the current data in its meta-analysis?.
Author Response
This manuscript needs some minor changes:
We would like to thank you for providing us with timely and valuable comments and suggestions to improve the quality of manuscript.
1) It is not clear if p-values in the tables are crude or corrected. Please indicate this.
The p values reported in the tables are adjusted by multiple logistic regression analysis. As reported in the text and in titles of the Tables 3 and 4, the p values were adjusted for sex and genomic ancestry. For Table 5 (corrected in the revised version of manuscript), genomic ancestry was used to stratify the sample, and the correction was based only in sex of the patients.
2) There is no mentioned to results for rs59902 by recessive model (significant p-value) in Results or Discussion.
Thank you for this very important observation. As highlighted, the variant rs599201 yielded association with NSCL±P only in the recessive model at a p value of 0.01, which is in the limit adopted in this study after applying the Bonferroni correction for multiple tests. Although this can be a true association, which deserves attention, this can also represent a spurious result due to the sample size, and multiple comparisons. With respect, we have limited the discussion to avoid speculation, focusing on the most consistent and relevant finding of the study, which involves the association of rs642961 polymorphism with the increased risk of NSCL±P.
3) Results within Table 5 are confused because showed association separated by ancestry origin but also adjusted by ancestry. In my opinion the authors must decide to segregate or to adjust.
Thank you for your important observation, and we are sorry for our mistake. The genomic ancestry was used to stratify the sample, and the correction was based only in sex of the patients. This was corrected in the revised version of manuscript.
4) Why the authors did not include the current data in its meta-analysis?.
The meta-analysis was initially performed to sample size estimation required for the study. However, during the writing of the discussion, we used the results to support our observation that the limited sample size, i.e., lack of magnitude to determinate the association, may have been the reason of the negative associations between rs642961 and NSCL±P in the previous studies based in the Brazilian population. It is important to consider that the results of the meta-analysis were quite similar to those we report in the current study individually. Moreover, the present sample is almost 5 times larger than the largest sample of the previous studies, and its inclusion would unbalance the weight of the articles, with important influences on the overall result of the meta-analysis.
Reviewer 3 Report
Comments and Suggestions for Authors
Beneath an overly elaborate title, I encountered a mediocre article that lacks sufficient novelty to warrant publication. The update to Zucchero et al. (2004) is rather weak. While it is beneficial to update such an old paper, this should be clearly stated. The excessive transmission of valine has not been addressed, despite the notion of highly significant results for some individual populations from South America and Asia.It is amazing how this paper connects with rs642961 in NSCL±P, as frequently replicated in Asian and European populations, but not in South America.The paper by Vieira et al. (2007) is not acknowledged at all, which seems to be a major flaw.
To ensure that authors are not left empty-handed, please note that subsections of the abstract do not have consecutive titles.
The term "IRF6" should be used in full when mentioned as a keyword.
In the introduction, orofacial malformations are attributed to the incomplete fusion of facial processes, while cleft palate openings occur due to the failure of the palatal shelves to fuse. It is important for authors to maintain consistency in their descriptions. Additionally, introducing a logical order in this initial paragraph could enhance clarity. For instance, when mentioning the processes for the first time, it would be helpful to provide their names.
Author Response
Reviewer was very disrespectful and aggressive, and provided useless comments. Do not deserve our attention.
Round 2
Reviewer 1 Report
Comments and Suggestions for Authors
The Authors have replied quite successfully to all points raised and modified their manuscript accordingly.
Reviewer 3 Report
Comments and Suggestions for Authors
I believe the author's response indicates that my comments were overlooked. Instead of being rude or litigious, they could have genuinely considered my input.